# The Role of Vitamin D and Vitamin D Receptor Gene Polymorphisms in the Course of Inflammatory Bowel Disease in Children

**DOI:** 10.3390/nu16142261

**Published:** 2024-07-13

**Authors:** Karolina Śledzińska, Anna Kloska, Joanna Jakóbkiewicz-Banecka, Piotr Landowski, Aleksandra Oppmann, Stephen Wilczynski, Agnieszka Zagierska, Barbara Kamińska, Michał A. Żmijewski, Anna Liberek

**Affiliations:** 1Department of Paediatrics, Haematology and Oncology, Medical University of Gdansk, 80-210 Gdansk, Poland; 2Department of Medical Biology and Genetics, University of Gdańsk, 80-308 Gdansk, Poland; anna.kloska@ug.edu.pl (A.K.); joanna.jakobkiewicz-banecka@ug.edu.pl (J.J.-B.); 3Department of Paediatrics, Gastroenterology, Nutrition and Allergology, Medical University of Gdansk, 80-210 Gdansk, Poland; piotr.landowski@gumed.edu.pl (P.L.); agnieszka.zagierska@gumed.edu.pl (A.Z.); bkam@gumed.edu.pl (B.K.); 4Department of Histology, Medical University of Gdansk, 80-210 Gdansk, Poland; ahc.oppmann@gumed.edu.pl (A.O.); swwilcz@gumed.edu.pl (S.W.); 5Department of Pediatrics, St. Adalbert Hospital, Copernicus PL Ltd., 80-462 Gdansk, Poland; anna.liberek@gumed.edu.pl

**Keywords:** inflammatory bowel disease, vitamin D receptor, *VDR*, polymorphism, vitamin D

## Abstract

**Background:** The etiopathogenesis of inflammatory bowel disease (IBD) is still unclear. Prior studies suggest genetic components that may influence the incidence and severity of the disease. Additionally, it was shown that low levels of serum vitamin D may have an impact on the clinical course of the disease due to its effect on the immunological system. **Methods:** We aimed to investigate the correlation between the incidence of vitamin D receptor (*VDR*) gene polymorphisms (rs11568820, rs10735810, rs1544410, rs7975232, and rs731236, commonly described as Cdx2, FokI, Bsm, ApaI, and TaqI, respectively) and vitamin D concentration with the clinical course of IBD (disease activity, extent of the intestinal lesions). Data were obtained from 62 patients with IBD (34 with Crohn’s disease, 28 with ulcerative colitis), aged 3–18 years, and compared with controls (N = 47), aged 8–18 years. **Results:** Although there was no difference in the incidence of individual genotypes between the study groups (IBD, C) in all the polymorphisms examined, we described a significant increase in the chance of developing IBD for heterozygotes of Cdx2 (OR: 2.3, 95% CI 0.88–6.18, *p* = 0.04) and BsmI (OR: 2.07, 95% CI 0.89–4.82, *p* = 0.048) polymorphisms. The mean serum 25OHD level in patients with IBD was significantly higher compared with the controls (19.87 ng/mL vs. 16.07 ng/mL; *p* = 0.03); however, it was still below optimal (>30 ng/mL). Furthermore, a significant correlation was found between vitamin D level and TaqI in patients with IBD (*p* = 0.025) and patients with CD (*p* = 0.03), as well as with the BsmI polymorphism in patients with IBD (*p* = 0.04) and patients with CD (*p* = 0.04). A significant correlation was described between the degree of disease activity and genotypes for the FokI polymorphism in patients with UC (*p* = 0.027) and between the category of endoscopic lesions and genotypes for the Cdx2 polymorphism also in patients with UC (*p* = 0.046). **Conclusions:** The results suggest a potential correlation of *VDR* gene polymorphism with the chance of developing IBD, and the clinical course of the disease requires further studies in larger group of patients. Vitamin D supplementation should be recommended in both children with inflammatory bowel disease and in healthy peers.

## 1. Introduction

The etiopathogenesis of inflammatory bowel disease (IBD) is multifactorial and has not yet been fully understood. It is believed that genetic, immunological, and environmental factors are involved in the development of chronic inflammatory lesions in the intestine [1]. The increased incidence of IBD among family members indicates the role of genetic factors. Crohn’s disease (CD) studies in Europe have shown a concordance rate of 30–50% in monozygotic twins and 10% in dizygotic twins [2,3,4]. Interestingly, Halfvarson et al. showed a similar localization of the disease in the intestine in most twins suffering from CD. The genetic component is less unequivocal in patients with ulcerative colitis (UC) [2]. The incidence of UC in monozygotic twins is about 16% and about 4% in dizygotic twins [2,3,4]. Comprehensive human genome-wide association studies (GWAS) have identified loci of more than a hundred IBD susceptibility genes, mainly within chromosomes 1, 5, 6, 12, 14, 16, and 19 [5,6,7]. More recent studies have shown up to 200 loci [8,9,10,11].

One of the candidate genes is the *VDR* gene encoding the vitamin D receptor located on the long arm of chromosome 12 (12q13.11). It consists of 11 exons; exons 2 and 3 encode the DNA binding domain, while exons 7, 8, and 9 participate in encoding the vitamin D binding domain [12]. VDR is commonly found in almost all the nucleated cells of the body. The active form of vitamin D after binding to the receptor induces direct genomic action by regulating the expression of about 3000 VDR-related genes, which accounts for about 10% of the human genome, but the effect strongly depends on the cell type and conditions [13].

More than 60 polymorphisms of the *VDR* gene have been described so far, most of which concern the promotor region, exons 2–9, and the 3′UTR region [14]. Their relationship to the occurrence of various diseases is still under investigation. Initially, researchers focused on the association of *VDR* gene polymorphism with bone metabolism parameters, mainly with osteoporosis [14]. In subsequent years, the association of *VDR* gene polymorphism with e.g., cancer or immunological disorders has been extensively studied [15,16]. Functional polymorphisms of the *VDR* gene include single nucleotide polymorphisms near the 3′UTR end recognized by endonucleases: ApaI, BsmI, and TaqI. BsmI (G/A) (rs1544410) and ApaI (G/T) (rs7975232) polymorphisms are located in introns between exons 8 and 9, while the TaqI (T/C) polymorphism (rs731236) is located in exon 9. These polymorphisms may affect mRNA stability and, consequently, the translation of VDR mRNA [17]. Another functional polymorphism is the FokI (C/T) polymorphism (rs10735810) located in exon 2, whose product exhibits greater biological activity [17]. The Cdx-2 polymorphism of the *VDR* gene (A/G) (rs11568820) has also been described, leading to reduced *VDR* expression in the intestine and, consequently, reduced calcium absorption [18]. Research on the frequency of the incidence of individual genotypes of *VDR* gene polymorphisms started over a decade ago, but the conclusions have remained ambiguous [19,20,21].

The aim of our study was to evaluate vitamin D status and *VDR* gene polymorphisms in the population of pediatric patients with IBD compared with controls to indicate the *VDR* gene polymorphisms predisposing to IBD and determining the clinical course of the disease and vitamin D status.

## 2. Materials and Method

### 2.1. Patients and Controls

The study involved 109 children, including 62 patients with IBD enrolled in the Department of Paediatrics, Gastroenterology, Hepatology and Paediatric Nutrition, Medical University of Gdańsk (MUG, Gdańsk, Poland), and 47 patients defined as the control/comparative group (C) enrolled in the Department of Paediatrics, Gastroenterology, Hepatology and Paediatric Nutrition, MUG and the Paediatric Emergency Department, COPERNICUS Hospital (Gdańsk, Poland). Table 1 shows the detailed characteristics of the study groups. The criterion for qualifying to the study group was the diagnosis of IBD, which was based on clinical presentation, the endoscopic and histopathological assessment of colon mucosa, and laboratory test results according to the PORTO criteria. Any inflammatory, infectious, immunological, nutritional, immunological, and cancer disorders during the 6 months prior to recruitment for this study were excluded in the control group.

Written informed consent was obtained from the parents or legal guardians of children and patients who were younger than 18 years of age. The study was approved by the Independent Bioethics Commission for Research at Medical University of Gdansk (No. 272/2011).

Past medical history was taken, and a thorough physical examination was conducted with laboratory tests performed (vitamin D concentration, calcium–phosphate balance parameters, inflammatory parameters). The qualification of both endoscopic and histological evaluations of the intestinal mucosa in patients with IBD was based on the PORTO criteria [22]. The Pediatric Ulcerative Colitis Activity Index (PUCAI) was used to assess the degree of severity of UC and the Pediatric Crohn’s Disease Activity Index (PCDAI) for patients with CD was assessed [23,24].

### 2.2. Serum Concentration of 25OHD

The concentration of 25OHD was determined using the immunochemiluminescence method using the DiaSorin Liaison XL analyzer (DiaSorin S.p.A., Saluggia VC, Italy; chemiluminescent immunoassay (CLIA)).

### 2.3. Analyses of VDR Gene Polymorphisms

Genomic DNA was isolated from whole blood samples using the QIAamp DNA Blood Mini Kit (QIAGEN, Germantown, MD, USA). The analysis of *VDR* gene polymorphisms:rs11568820 (Cdx2)—NG_008731.1(NM_000376.3):c.-3891G>A,rs1544410 (BsmI)—NG_008731.1(NM_000376.3):c.1024+283G>A,rs7975232 (ApaI)—NG_008731.1(NM_000376.3):c.1025-49G>T,rs731236 (TaqI)—NG_008731.1(NM_000376.3):c.1056T>C (p.Ile352=),rs10735810 (FokI)—NG_008731.1(NM_000376.3):c.125T>C (p.Met=),


was performed according to Lins et al. [25], with minor modifications.


In brief, fragments of the *VDR* gene were PCR-amplified in a multiplex reaction using the previously described PCR primers [25] and Taq DNA polymerase (Roche, Basel, Switzerland). Following the enzymatic purification of the PCR products by incubation with exonuclease I (ExoI) and shrimp alkaline phosphatase (SAP) (both from USB Products Affymetrix, Inc., Santa Clara, CA, USA), the minisequencing reaction was performed using a SNaPshot^®^ Multiplex Kit (Applied Biosystems, Waltham, MA, USA) and the previously described single-base extension primers [25]. Sequencing products were enzymatically purified using SAP treatment, mixed with Hi-Di formamide (Applied Biosystems, Waltham, MA, USA), and denatured and separated using capillary electrophoresis on a ABI PRISM^®^ 310 Genetic Analyzer (Applied Biosystems, Waltham, MA, USA) in denaturing conditions (60 °C) using the POP-4 Polymer (Applied Biosystems, Waltham, MA, USA) as the separation matrix and the GeneScan™ 120 LIZ^®^ Size Standard (Applied Biosystems, Waltham, MA, USA) as the internal size standard. Genotypes were determined by analysis of electropherograms in Peak Scanner Software v1.0 (Applied Biosystems, Waltham, MA, USA). The overall success rate for the detection of all the *VDR* gene polymorphisms in all the samples was 100%. It was achieved by a combination of detailed analyses of electropherograms and microsequencing.

### 2.4. Statistics

Statistica 8.0 [StatSoft, 1984–2011, Tulsa, OK, USA] was used for the statistical calculations. The basic characteristics of the quantitative variables were established—arithmetic mean, median, minimum, maximum, and standard deviation. Each quantitative variable was subjected to the Shapiro–Wilk normality test. Differences were statistically significant at *p* < 0.05 (t-student test or Mann–Whitney–Wilcoxon test). The degree of mutual interdependence was determined by the Pearson or Spearman correlation coefficient. Qualitative variables were sorted and analyzed using the chi-square test with or without Yates correction (by sample size). GraphPad Prism 5 (GraphPad Software, Inc., San Diego, CA, USA) was used to graph the data.

## 3. Results

### 3.1. Analysis of VDR Gene Polymorphisms

To study the influence of *VDR* gene polymorphisms in the course of inflammatory bowel disease in children, the group of 109 children, including 62 patients, were selected (Table 1). The frequencies of the individual alleles of the *VDR* gene polymorphisms are shown in Table 2.

The frequency of genotypes for *VDR* gene polymorphisms was consistent with the Hardy–Weinberg equilibrium in both the investigated and comparative groups, apart from the rs1544410 (BsmI) and rs731236 (TaqI) polymorphisms in the comparative group (Table 3).

There was no statistically significant difference in the incidence of individual genotypes between the study groups (IBD, C) in all the polymorphisms examined—rs11568820 (Cdx2) (*p* = 0.13), rs10735810 (FokI) (*p* = 0.59), rs1544410 (BsmI) (*p* = 0.2), rs7975232 (ApaI) (*p* = 0.25), and rs731236 (TaqI) (*p* = 0.15), and the subgroups (CD, UC, C) in all the polymorphisms examined—rs11568820 (Cdx2) (*p* = 0.33), rs10735810 (FokI) (*p* = 0.33), rs1544410 (BsmI) (*p* = 0.4), rs7975232 (ApaI) (*p* = 0.45), and rs731236 (TaqI) (*p* = 0.38).

To extend the statistical analyses of the impact of *VDR* polymorphism on the development of IBD, the risk ratio (RR) and observed risk (OR) ratio values and 95% confidence intervals (CIs) were calculated (Table 4). It was shown that there was a statistically significant increase in the chance of developing IBD in heterozygotes for the rs11568820 (Cdx2) polymorphism (OR: 2.3, 95% CI [0.88, 6.18], *p* = 0.04), in heterozygotes for the rs1544410 (BsmI) polymorphism (OR: 2.07, 95% CI [0.89, 4.82], *p* = 0.048), and in heterozygotes for the rs731236 (TaqI) polymorphism (OR: 2.18, 95% CI [0.92, 5.2], *p* = 0.05). Interestingly, the opposite effect, with an increased chance of developing IBD, was observed in GG homozygotes for the rs7975232 (ApaI) polymorphism (OR: 0.47, 95% CI [0.21, 1.04], *p* = 0.05) and TT homozygotes for the rs731236 (TaqI) polymorphism (OR: 0.47, 95% CI [0.21, 1.03], *p* = 0.04). In other cases, there was no evidence of an increased chance of developing IBD or the differences observed were not statistically significant. 

### 3.2. Serum Vitamin D Concentration and Correlation with VDR Gene Polymorphisms

The serum 25-hydroxyvitamin D (25OHD) concentration in the IBD group was significantly higher compared with the control group (19.87 ± 10.15 ng/mL vs. 16.07 ± 6.35 ng/mL, respectively, *p* = 0.03). In the analyzed subgroups, the serum vitamin D level in patients with CD was higher than in the controls (20.21 ± 9.6 ng/mL vs. 16.07 ± 6.35 ng/mL; *p* = 0.04), but it did not differ from controls in patients with UC (19.45 ± 10.9 ng/mL vs. 16.07 ± 6.35 ng/mL; *p* = 0.1).

In the majority of patients with IBD (59.7%) and the comparative group (80.85%), vitamin D deficiency was identified (25OHD < 20 ng/mL); a suboptimal concentration (20–30 ng/mL) was found in 27.4% and 15% of children, respectively. Normal vitamin D levels (30–50 ng/mL) were detected only in 9.7% of patients with IBD and 4.25% of children in the comparative group. Two children with IBD had vitamin D levels above 50 ng/mL.

Analyzing the total number of study participants (N = 109, IBD + C), it was shown that the mean vitamin D level was slightly lower in girls as compared with boys (17.25 ng/mL ± 8.45 vs. 18.95 ng/mL ± 9.2, respectively, *p* = 0.322). There was no statistically significant correlation between the vitamin D level with the age of all the children studied (IBD + C) (R = −0.004, *p* = 0.966).

A statistically significant correlation was found between the vitamin D level and the rs731236 (TaqI) polymorphism in patients with IBD (*p* = 0.025) and patients with CD (*p* = 0.03). Patients with the TT genotype (homozygote-dominant) had significantly higher serum vitamin D levels as compared with the CC genotype (recessive homozygotes) (Figure 1).

Additionally, statistically significant correlation was found between vitamin D level and rs1544410 (BsmI) polymorphism in IBD (*p* = 0.04) and CD (*p* = 0.04) patients. Patients with GG genotype (homozygote dominant) had significantly higher serum vitamin D levels as compared to AA genotype (recessive homozygotes) (Figure 2).

### 3.3. Disease Activity in Patients with IBD and Correlation with VDR Gene Polymorphisms

Based on the number of points calculated from the PCDAI/PUCAI scale, patients were enrolled in one of the disease activity groups (remission, mild, moderate, severe). Among patients with UC, there were none with the severe degree of the disease (Table 5).

A statistically significant correlation was found between the degree of disease activity and genotypes for the rs10735810 (FokI) polymorphism only in patients with UC (*p* = 0.027) (Figure 3).

### 3.4. Extent of the Disease and Correlation with VDR Gene Polymorphisms

According to the Paris classification, patients with IBD were divided into groups depending on the extent of pathological lesions found in the gastrointestinal tract during the endoscopic examination [26]. Among patients with CD, the most common category was L2 (colonic)—62%, then L3 (ileocolonic)—32% and L1 (distal 1/3 ileum)—6%. None of the patients with CD had upper gastrointestinal involvement (L4). Among patients with UC, the dominant category was E3 (extensive)—43%, E2 (left-sided ulcerative colitis) was found less frequently—21%, then E1 (proctitis)—18% and E4 (pancolitis)—18%. A statistically significant correlation was found between the category of endoscopic lesions and genotypes for the rs11568820 (Cdx2) polymorphism only in patients with UC (*p* = 0.046) (Figure 4).

## 4. Discussion

In the present study, rs11568820 (Cdx2), rs10735810 (FokI), rs1544410 (BsmI), rs7975232 (ApaI), and rs731236 (TaqI) polymorphisms of the vitamin D receptor gene were analyzed in children with IBD and a comparative group. There was no statistically significant difference in the incidence of individual genotypes between groups (IBD, C) or subgroups (CD, UC, C) for all polymorphisms studied. At the same time, the chance of developing IBD was 2.3-fold higher for the genotype GA of the rs11568820 (Cdx2) polymorphism and 2-fold higher for the genotype GA of the rs1544410 (BsmI) polymorphism.

Recently, the *VDR* gene polymorphism in IBD was subjected to a meta-analysis. The first meta-analysis by Xue et al. showed the relationship between the genotype CC of the rs731236 (TaqI) polymorphism and the prevalence of CD among Europeans [20], but of CC genotype were similar in patients with IBD and the control group. The mechanism of this phenomenon is difficult to explain [20]. In the rs731236 (TaqI) polymorphism, thymine is substituted by cytosine in exon 9 from the 3′-end of the *VDR* gene, but it does not result in an amino acid change as both codons encode isoleucine. However, the gene region, located at the 3′-end, close to this polymorphic site, is involved in regulation of expression of the gene affecting mRNA stability [20]. Another hypothesis that should also be considered claims that the CC genotype of the rs731236 (TaqI) polymorphism may exist in a linkage disequilibrium with other factors that increase the risk of CD [20]. Interestingly, in our study, the CC genotype of the rs731236 (TaqI) polymorphism was found to be associated with a low 25(OH)D serum level in both groups. The meta-analyses also pointed out the increased incidence of the CC genotype of the rs10735810 (FokI) polymorphism among Asians. However, this effect was not observed in our European population. This polymorphic site is located in the 5′-end of exon 2 of the *VDR* gene. The substitution of thymine to cytosine in exon 2 leads to the synthesis of a protein that is three amino acids longer and has a lower activation capacity for VDR-target genes [20]. Lower VDR activity may result in an increased susceptibility to IBD. It seems that the carrier of the recessive T allele (TT, GT genotypes) of the rs7975232 (ApaI) polymorphism has a protective effect and is associated with a reduced risk of CD [27]. The rs7975232 (ApaI) polymorphism is located in the 3′-end of intron 8 of the *VDR* gene; however, the exact effect of the guanine-to-thymine substitution is not known.

The second meta-analysis by Wang et al. describes the differences in the incidence of particular polymorphisms depending on the origin of the patients (place of residence, race, ethnic group) [21]. The rs731236 (TaqI) polymorphism T allele was found in 55.04% to 65.96% of Caucasians and as much as 94.35% of people in eastern Asia [21]. Interestingly, this is in agreement with our data (69%). Pei et al.’s study comparing 218 Han Chinese UC cases with 251 control subjects concluded that the frequency of the BB genotype of the rs1544410 (BsmI) polymorphism was increased among the case group. Furthermore, the rs1544410 (BsmI) frequency of the polymorphic allele (B) was associated with Han Chinese UC cases. Despite this, there was no association found between this genotype and allele frequencies and the clinical course of UC among cases [28]. Perhaps this divergence is influenced by different genetic origins or different environmental factors, or there are different patterns of imbalance. In addition, Wang et al.’s study showed that the rs7975232 (ApaI) polymorphism was associated with an increased risk of CD, and the rs731236 (TaqI) polymorphism showed a protective effect and reduced the risk of CD [21]. The rs7975232 (ApaI) and rs1544410 (BsmI) polymorphisms are considered as “silent” single nucleotide polymorphisms (SNPs) that do not alter the amino acid sequence encoding the protein but may affect gene expression by regulating mRNA stability [29].

The rs11568820 (Cdx2) polymorphism was the first to be described among women living in Japan, but later it was also shown to be present among Caucasian and other ethnic groups [18]. In this polymorphic site, guanine is substituted by adenine in the promoter region, exactly at the specific intestinal transcription-factor-binding site called Cdx2 [30]. Allele A was described as more active leading to the binding of the Cdx2 transcription factor, thereby increasing its transcriptional activity [31]. Consequently, this may result in increased *VDR* gene expression in the intestine and increased mineral bone density through increased calcium absorption in the gastrointestinal tract [31].

A study conducted by Hughes et al. in a large Caucasian group of 660 patients with IBD and 699 healthy individuals found no link between FokI, BsmI, ApaI, and TaqI polymorphisms of the *VDR* gene with the risk of IBD [32]. Simmons et al. (UK) described twice times higher the risk of CD in patients with a homozygote recessive genotype (CC) of the rs731236 (TaqI) polymorphism [33]. This is consistent with our study.

Children with IBD are a specific risk group for vitamin D deficiency [34]. In the course of the disease, it is typical to develop impaired absorption, enteropathy with secondary protein loss, and inflammation in the intestine. Very often, due to hospitalization or temporary immobilization, patients also experience reduced exposure to sunlight and reduced vitamin D intake. Particular attention should be paid to the problem of hypovitaminosis D in dark-skinned children who live in northern latitudes, with a severe course of the disease and early onset, low albumin level, lower BMI, and abnormalities observed in the densitometry examination. An additional important risk factor is the involvement of the upper gastrointestinal tract in the course of the disease [35,36].

In the present study, it was interesting to observe a significantly lower 25OHD concentration in the comparative group (16.07 ng/mL) as compared with the IBD group (19.87 ng/mL). One of the reasons may be the widespread deficiency of vitamin D in healthy children, especially teenagers. On the other hand, it is known that children with IBD remain very often under more careful supervision of their caregivers and medical staff compared with their healthy peers (teens). It is known that “non-compliance” is a significant problem in this age group [37]. Most importantly, in neither group (IBD and control), did the mean vitamin D level reach 20 ng/mL, indicating a significant deficiency in this important vitamin [34].

The NHANES study of 6275 children and adolescents aged 1 to 21 years revealed a vitamin D deficiency (<15 ng/mL) in 9% of subjects and a suboptimal concentration (15–29 ng/mL) in 60% of subjects. These data extrapolated to the entire population of American children show that as many as 60 million of them have vitamin D deficiency [38].

Most studies assessing vitamin D levels in the pediatric population in Poland have been performed in the youngest (neonates, infants) and data on adolescents are sparse [39]. In a study conducted by Karczmarewicz et al. in children aged 2–19 years, the mean concentration of 25OHD was 17.49 ng/mL, and an inverse proportional relationship of 25OHD concentration to age was observed (24.06 ng/mL in 2–4 years old children and 12.84 ng/mL among the youth) [40]. Śledzińska et al.’s study focusing on children with IBD of average age 14.4 concluded that despite 25OHD levels being higher in children with IBD than the control group (18.1 ng/ML as compared with 15.5 ng/mL), there was no correlation between IBD duration and 25OHD levels [41].

Our analysis of the association of *VDR* gene polymorphisms with vitamin D levels in the study groups and subgroups showed a statistically significant correlation of lower vitamin D levels in patients with IBD and patients with CD with the CC genotype of the rs731236 (TaqI) polymorphism over dominant homozygotes (TT genotype). A similar correlation was observed in patients with IBD and patients with CD with the AA genotype of the rs1544410 (BsmI) polymorphism over dominant homozygotes (GG genotype). In addition, a statistically significant correlation was found between the degree of disease activity and the rs10735810 (FokI) polymorphism in patients with UC and between the category of endoscopic lesions and the rs11568820 (Cdx2) polymorphism also in patients with UC. Up to date, studies on the association of *VDR* gene polymorphisms with disease activity, its location, or selected laboratory parameters in patients with IBD are few. In a study conducted in New Zealand, a country with one of the highest rates of incidence of IBD, lower vitamin D levels were reported in patients with CD compared with the control group. Low vitamin D levels correlated with age and little exposure to sunlight. In addition, the correlation of low concentrations of 25OHD with the rs731236 (TaqI) polymorphism of the *VDR* gene in patients with CD and the rs7975232 (ApaI) polymorphism of the *VDR* gene in control subjects was observed [42]. Also, Hustmeyer et al., demonstrated a rs731236 (TaqI) polymorphism association with low vitamin D concentration in healthy populations [43]. It appears that the CC genotype of this polymorphism may be associated with a reduced level of the vitamin D receptor mRNA [44]. This may result in the decrease in active serum vitamin D concentration and a reduction in its inhibitory effect on IL-12 [45]. The involvement of the immune system manifests itself as an increased Th1-dependent reaction leading to an increased susceptibility to CD. An analysis by Naderi et al. of 230 Iranians with IBD showed a higher prevalence of the CC genotype of the rs10735810 (FokI) polymorphism, but no association was observed between the occurrence of the polymorphism and the extent of the disease in the gastrointestinal tract, the endoscopic location of the disease, or its activity [46]. Martin et al. analyzed the association between the rs731236 (TaqI) polymorphism, in 188 patients with IBD and 115 controls, and the disease phenotype showed an increased frequency of the CC genotype in patients with stenosis and fistulas (*p* = 0.04), while the presence of the TT genotype was protective (*p* = 0.006) [47]. Whereas Xia et al. demonstrated that the presence of mutant alleles (TT, GT genotype) of rs7975232 (ApaI) polymorphism and vitamin D deficiency correlated with an increased risk of CD [48]. In addition, in patients with CD, vitamin D deficiency was associated with the presence of mutant alleles (TC+CC genotype) of the rs10735810 (FokI) polymorphism, the GT+TT genotype of the rs7975232 (ApaI) polymorphism, and the TC+CC genotypes of the rs731236 (TaqI) polymorphism. Decreased vitamin D level was also observed in patients who were homozygous for rs10735810 (FokI) ancestral alleles [49].

## 5. Conclusions

The results obtained in this study and the literature review may be the basis for further analysis. Further reports underline the role of the pleiotropic action of vitamin D, but there are still not enough properly designed randomized trials in patients with IBD, especially among pediatric patients. The limitation of this study was the lack of a direct bone-density analysis, which would enrich our knowledge of its relation to vitamin D levels in patients with IBD. It is important to note that multiple comparisons should be subjected to the Bonferroni correction. However, due to the relatively small study group and the potentially high variability of the tested *VDR* gene polymorphisms in the general population, the application of such a correction resulted in a complete loss of statistical significance of the presented data. Thus, further studies are required to validate the significance of the *VDR* gene polymorphism in the development of IBD.

It is worth adding that according to the latest recommendations, a vitamin D level below 20 ng/mL indicates its deficiency, while it is desirable to achieve a level of at least 30 ng/mL [34]. This indicates the need to implement appropriate supplementation in both groups.

## Figures and Tables

**Figure 1 nutrients-16-02261-f001:**
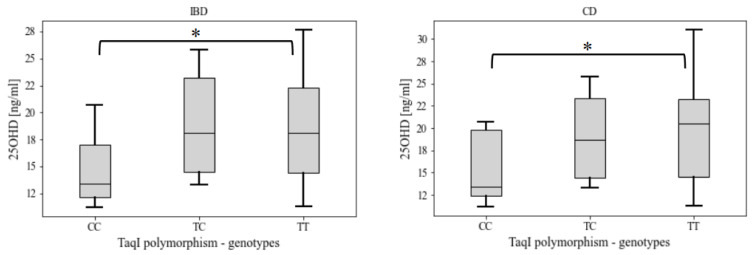
Serum 25OHD levels among genotypes of the rs731236 (TaqI) polymorphism in patients with IBD and patients with CD * statistical significance at *p* < 0.05. Data are presented as median with upper and lower quartile (box) and min–-max observed values (whiskers). IBD—inflammatory bowel disease, CD—Crohn’s disease.

**Figure 2 nutrients-16-02261-f002:**
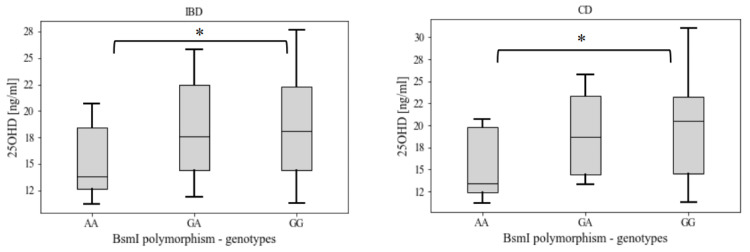
Serum 25OHD levels among genotypes of the rs1544410 (BsmI) polymorphism in patients with IBD and patients with CD. Data are presented as median with upper and lower quartile (box) and min–max observed values (whiskers). * statistical significance at *p* < 0.05. IBD—inflammatory bowel disease, CD—Crohn’s disease.

**Figure 3 nutrients-16-02261-f003:**
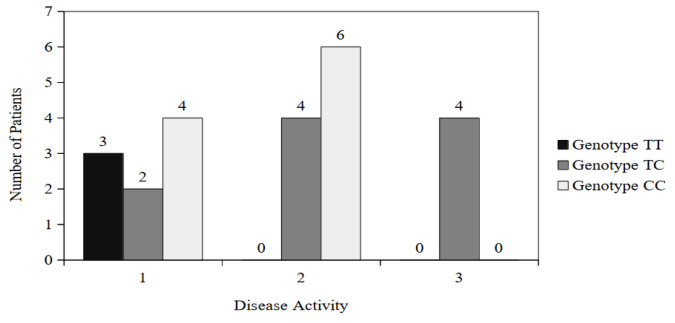
Genotype distribution for the rs10735810 (FokI) polymorphism among patients with UC from the different disease activity groups.

**Figure 4 nutrients-16-02261-f004:**
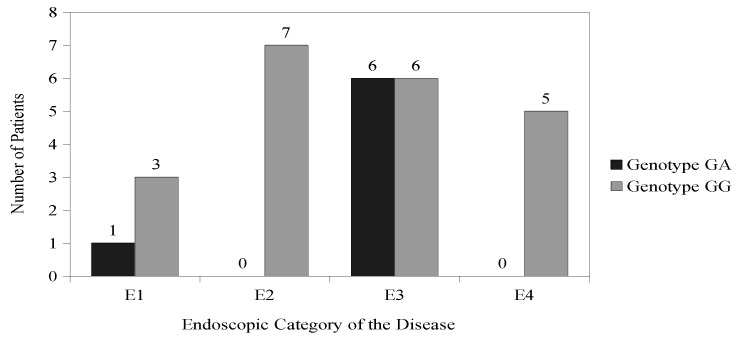
Genotype distribution for the rs11568820 (Cdx2) polymorphism among patients with UC with different endoscopic locations of the disease.

**Table 1 nutrients-16-02261-t001:** Age and gender of children in the IBD and control groups.

Patient Group	Number	Mean Age ± SD (Years)	Females	Males
C	47	13.97 ± 2.57	23	24
IBD	62	14.4 ± 3.01	23	39
CD	34	14.83 ± 2.17	10	24
UC	28	13.86 ± 3.11	13	15

IBD—inflammatory bowel disease, CD—Crohn’s disease, UC—ulcerative colitis, C—control, SD—standard deviation.

**Table 2 nutrients-16-02261-t002:** Allele frequencies (%) of *VDR* gene polymorphisms in the IBD group and the control (C) group.

Polymorphism	Common Name	Allele	Frequency (%)	
			IBD	C
rs11568820	Cdx2	G	53 (85)	42 (90)
		A	9 (15)	5 (10)
rs10735810	FokI	T	29 (47)	24 (52)
		C	33 (53)	23 (48)
rs1544410	BsmI	G	43 (69)	37 (78)
		A	19 (31)	10 (22)
rs7975232	ApaI	G	33 (54)	30 (64)
		T	29 (46)	17 (36)
rs731236	TaqI	T	43 (69)	37 (79)
		C	19 (31)	10 (21)

IBD—inflammatory bowel disease, C—control.

**Table 3 nutrients-16-02261-t003:** Genotype frequencies (%) of *VDR* gene polymorphisms in the IBD group and the control (C) group with χ^2^ results of the Hardy–Weinberg equilibrium testing. The result is significant at *p* < 0.05.

Polymorphism	Genotype	IBD			C		
		Number of Patients (%)	χ^2^	*p* Value	Number of Patients (%)	χ^2^	*p* Value
rs11568820 (Cdx2)	GG	44 (71)	1.79	0.18	39 (83)	0.92	0.34
	GA	18 (29)			7 (14.9)		
	AA	0 (0)			1 (2.1)		
rs10735810 (FokI)	TT	12 (19.4)	0.64	0.42	13 (27.7)	0.02	0.89
	TC	34 (54.8)			23 (48.9)		
	CC	16 (25.8)			11 (23.4)		
rs1544410 (BsmI)	GG	31 (50)	0.49	0.48	31 (66)	4.98	0.03 *
	GA	24 (38.7)			11 (23.4)		
	AA	7 (11.3)			5 (10.6)		
rs7975232 (ApaI)	GG	17 (27.4)	0.32	0.57	21 (44.7)	1.37	0.24
	GT	33 (53.2)			18 (38.3)		
	TT	12 (19.4)			8 (17)		
rs731236 (TaqI)	TT	31 (50)	1.21	0.27	32 (68.1)	6.26	0.01 *
	TC	23 (37.1)			10 (21.3)		
	CC	8 (12.9)			5 (10.6)		

IBD—inflammatory bowel disease, C—control, * *p* < 0.05.

**Table 4 nutrients-16-02261-t004:** Risk ratio, relative risk of *VDR* gene polymorphisms in the IBD in comparison with the control group.

Polymorphism	Genotype	Risk Ratio (RR)	Approximate 95% Confidence Interval (Koopman)	Approximate Power (for 5% Significance)	Observed Odds Ratio	Approximate 95% Confidence Interval (Woolf, Logit)	*p* (Fisher)
rs11568820 (Cdx2)	GG	0.767	0.57 to 1.11	22.14%	0.5	0.20 to 1.28	0.11
	GA	1.37	0.96 to 1.85	31.4%	2.34	0.88 to 6.18	0.04 *
	AA	0	0 to 1.41	<0.01%	0	0 to 29.56	NS
rs10735810 (FokI)	TT	1.11	0.80 to 1.56	6.13%	1.27	0.59 to 2.71	0.3
	TC	0.81	0.49 to 1.19	12.24%	0.63	0.26 to 1.54	0.2
	CC	1.06	0.70 to 1.47	5.83%	1.14	0.47 to 2.75	0.4
rs1544410 (BsmI)	GG	0.76	0.55 to 1.05	30.88%	0.52	0.24 to 1.13	0.05 *
	GA	1.34	0.95 to 1.82	31.16%	2.07	0.89 to 4.82	0.048 *
	AA	1.03	0.55 to 1.52	5.04%	1.07	0.32 to 3.60	0.5
rs7975232 (ApaI)	GG	0.71	0.46 to 1.02	38.52%	0.47	0.21 to 1.04	0.04 *
	GT	1.26	0.91 to 1.76	21.71%	1.72	0.80 to 3.71	0.11
	TT	1.07	0.67 to 1.51	5.92%	1.17	0.44 to 3.14	0.4
rs731236 (TaqI)	TT	0.73	0.53 to 1.01	39.38%	0.47	0.21 to 1.03	0.04 *
	TC	1.36	0.97 to 1.84	34.12%	2.18	0.92 to 5.20	0.05 *
	CC	1.09	0.62 to 1.57	6.14%	1.24	0.38 to 4.08	0.4

NS—not significant, * *p* < 0.05.

**Table 5 nutrients-16-02261-t005:** Degree of disease activity in patients with CD and patients with UC.

Patient Subgroup	Degree of Disease Activity			
	1 Remission	2 Mild	3 Moderate	4 Severe
	Number of Patients/Frequency (%)			
CD	10 (29.41)	16 (47.06)	4 (11.76)	4 (11.76)
UC	12 (42.86)	13 (46.43)	3 (10.71)	0 (0)

## Data Availability

Data are contained within the article.

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
