# Peer review of "The Role of Vitamin D and Vitamin D Receptor Gene Polymorphisms in the Course of Inflammatory Bowel Disease in Children"

_nutrients, 2024, doi:10.3390/nu16142261_

Round 1

Reviewer 1 Report

Comments and Suggestions for Authors

It is an important topic to study on,  however, several questions need to be addressed  as well: 

1. In the method part: 

1) no description about the success rate of each VDR gene polymorphsims was detected.  (2.2 analysis of VDR gene polymorphisms, analysis should be replaced with analyses)

2) no descripton abouth the OR calculation and no table showed it. In my opion, the table 3 could be a supplementary file.  and  adding a table of how you calculated the OR and 95%CI. Which model was used to calculate the OR value, the additive model or recessive model or dominant model, what kind of variables were adjusted ?

3) Based on the current sample size,  how is the statistical power ?

2. Discussion part

1) should describe the SNP will increase or decrease the corresponding gene expression in mRNA or protein level, how it would impact on the disease and why it will impact the concentration of VitD.

2)  the exon change should be expressed in amino acid change, like PNPLA3 rs738409 (I148M)

Comments on the Quality of English Language

 Minor editing of English language required

Author Response

Thank you for all your comments, the answers for your convenience are placed in a file.

Reviewer 2 Report

Comments and Suggestions for Authors

The authors are attempting to establish a link between some VDR gene polymorphism and the risk of developing IBD.
Statistics is everything in this study as its conclusions rely essentially on the results of the statistical calculations - which, unfortunately, are faulty. Although multiple comparisons have been performed, no strategy for dealing with the multiple comparisons problem was employed and therefore the statistics in this article is only an exercise in p-hacking. For example, in Table 3 Bonferroni correction should have been applied and therefore the threshold for statistical significance should have been lowered to 0.05/(number of comparisons, namely 5) = 0.01 - consequently only the last result (that for rs731236 (TaqI)) is statistically significant (and only marginally). It is doubtful whether any of the results presented in the abstract would survive Bonferroni correction (the lowest p-value is 0.025 - there were most certainly more than 2 comparisons performed during the statistical analysis).
In the legend of figures 1 and 2 it is written "*statistical significance at =p<0.05." I can see no "*" in the figures.
For every statement / paragraph in the Results section the relevant table / figure should be cited. I do not know where are the results evoked in the paragraphs:
"There was no statistically significant difference in [...] rs731236 (TaqI) (p = 0.38). 147
and
"However, there was a statistically significant increase [...] observed were not statistically significant."

Author Response

Thank you for all your comments, the answers for your convenience are placed in a file
